# Factors Influencing the Chance of Dropout or Being at Risk of Dropout in Higher Education

Gabriella Pusztai [1] , Hajnalka Fényes [2] and Klára Kovács [1,*]

1   MTA-DE-Parent-Teacher Cooperation Research Group, Institute of Educational Studies and Cultural Management, University of Debrecen, 4032 Debrecen, Hungary
2   MTA-DE-Parent-Teacher Cooperation Research Group, Department of Sociology and Social Policy, University of Debrecen, 4032 Debrecen, Hungary
*   Correspondence: kovacs.klara@arts.unideb.hu

**Abstract:** The purpose of this paper is to establish what sociodemographic and institutional factors cause students to drop out, become uncertain about their intentions to obtain a degree, or confidently advance towards the fulfilment of their ambitions. Our analysis is based on the combined databases of large-sample questionnaire surveys carried out among former students who dropped out from higher education institutions in an eastern region of Hungary as well as those carried out among current students. In addition to bivariate methods, we conduct multinomial logistic regression analysis to explore how students' gender, social background, the funding of their training, willingness to do paid work alongside their studies, and relationships with academic staff and fellow students affect the chance of dropout, the risk of dropout, and persistence. In contrast to previous studies, which have mostly identified those at risk of dropping out of higher education and have primarily focused on the deficiencies of institutional integration, our novel results show that the actual dropout rate is at least as influenced by students' unfavourable social background as it is by institutional factors.

**Keywords:** dropout; Central and Eastern Europe; sociodemographic background; institutional integration

## 1. Introduction

Increasing the proportion of people with tertiary degrees, attracting more students to higher education, and retaining them are among the top priorities of European governments' educational policies. In our research questions, we ask how students who dropped out, who are at risk of dropout, and who are persistent differ in their main sociodemographic background variables, the funding of their training, their willingness to do paid work alongside their studies, and their embeddedness in the institution. Our research can compare students who did drop out with those at risk of dropping out and those who persist with their studies, thus revealing further differences and identifying factors that increase or may mitigate student dropout.

In the United States and Western Europe, the expansion of higher education was already ongoing in the 1960s, but in Central and Eastern European (CEE) countries under socialism, governments sought to keep the higher education enrolment rate below 10% [1,2]. As a result, in the Western world, while the children of those who had enrolled in higher education in the first wave of expansion were already attending university in the 1990s, the share of first-generation students in CEE is still high. These students and their families find it difficult to cover the costs associated with higher education (accommodation, meals, transportation, and study equipment). Furthermore, the first-generation students' parents are often unable to provide their children with the relevant intellectual experience and background [3].

Under socialism, the few students who were able to enrol in higher education could study free of charge; However, today, many students must pay tuition fees. Students can only study tuition-free for 12 semesters, provided that their secondary school-leaving

examination results and university grades reach the statutory threshold. Tuition fees vary across study programs, but in the most prestigious fields of study, they are well above the EU average. For those who remain or drop below the academic threshold and thus receive no state funding, finding the money to pay tuition fees is an immense challenge. Slow credit acquisition and deteriorating exam results lead to the loss of exemption from tuition and the significant cost of tuition often forces students to take on disproportionately burdensome employment in parallel with their studies [4].

In Hungary, the sudden expansion of higher education at the turn of the millennium led to a significant increase in capacity, a relative abundance of institutions, and competition for students. The Hungarian higher education system did not materially revise the content of the curriculum at the time of the Bologna reform: the curricula of the previous five-year programs, which were often too theoretical, were condensed into three-year undergraduate programs. Furthermore, certain undergraduate qualifications do not enable employment in relevant fields. An additional problem is that higher education institutions often fail to keep pace with the changing composition of the student population and fail to employ outdated pedagogical methods. This makes it difficult to retain students who are not particularly engaged, motivated, and ambitious [5].

Student dropout is defined in different ways in the literature, depending on whether it is considered from the perspective of the higher education institution (attrition) or the individual (dropout). In primary and secondary education, the term early school leaving from education and training is used instead of dropout but our article focuses on the higher education level and the individual perspective. Seidman [6] considers dropout to be the interruption of one's studies and contrasts it with persistent advancement, whereby students are committed to obtaining a degree and make the required effort to do so [7,8].

The causes of dropout have been searched for since the expansion of higher education began [9]. In Europe, research started well after it did in the USA [9,10]. The Bologna reform divided the relatively long years of study into cycles with a view to accelerating graduation and making it attainable to a wider range of students [2]. The structural reform, however, did not lead to less attrition and, even in the 2010s, understanding the phenomenon remained one of the main challenges of higher education research. Researchers call for investigation that takes into consideration the national characteristics of higher education systems, which have a strong effect on attrition [11,12]. The main difference between the national higher education systems is that how selective they are, when the student enrols in and graduates from higher education [12]. Another difference is how regulations and the curriculum hinder the advancement and career correction opportunities of students. Furthermore, there are differences in tuition payment systems, while student grants and benefits are neither widely available nor sufficient to live on [13–15].

To ground our analysis, we outline the potential explanatory factors for dropout and the most important findings of our research into dropout so far in this paper. After giving the methodological background, we elaborate on the differences between students who dropped out, those who are at risk of dropout, and those who are persistent. We used multivariate statistical methods to determine the effects of the chief explanatory factors.

## 2. Literature Review

### 2.1. The Trends and Reasons of Student Dropout

There are approximately 17.5 million students completing their higher education in EU countries, a significant proportion of whom will never graduate [12]. It is virtually impossible to draw comparisons between countries, not only because regulations of tertiary studies differ but also because methods of measuring dropout are inconsistent (either the student population of one year group is tracked throughout their studies or the ratio of students enrolling and leaving higher education in a given year is registered). Regardless of measurement methods, dropout rates can vary greatly across countries. For example, in the UK, over 70 percent of bachelor students graduate within the theoretical duration of the program, but in the Netherlands, Belgium, and Austria, less than 30 percent do

so [16]. In Central and Eastern Europe, the proportion of students who do not complete their studies is higher than the EU average [17–20]. Regarding the proportion of students who completed their tertiary studies, Hungary ranked in the middle of EU countries in the first two decades of the 21st century [12].

Several researchers have differentiated between dropout students based on their social background, career choice, and institutional experiences [21–26]. A Central European research project conducted in 2018 differentiated between four types of dropout students, applying cluster analysis [27]. The first type includes students who drop out for financial reasons: the costs of their studies (transportation, daily expenses, accommodation, textbooks and school supplies, IT equipment, etc.) are high while student benefits (subsidized dormitories, student grants, and family allowance) are not available to everyone. They are students of low social status who do not receive any support from their families and take on too much paid work to successfully pursue their studies [28]. The second type consists of students who have academic difficulties and suffer from the lack of support from the institution. They attribute their failures to the inflexibility of the curriculum, the inappropriate flow of information, an institutional environment that hinders efficient learning and student independence, indifferent or incompetent lecturer(s), and office staff who withhold information [22,24,26]. Kuh and colleagues also consider that one of the most important factors of dropout is when student(s) do not perceive any support to overcome their social and academic problems [29].

Conspicuously, there are many whose performance deteriorates despite their promising results at the beginning of their studies. Those who took advanced school-leaving exams, have advanced language certificates, and ranked high in competitions are overrepresented in this cluster. The third type includes those who become disillusioned with higher education or, having made a bad career decision, end up in a degree program or institution that is unsuitable for them, or become disappointed by the outdated study materials and style of education [21]. In this situation, students who come from favourable backgrounds usually make another attempt at another degree program [30]. The fourth type comprises dropout students who are in the most hopeless situation, having both academic and financial difficulties. Most of them have nobody from faculty with whom their problems can be discussed after the lectures. Of the other students, less than half had the opportunity to do so. Although dropout students are a diverse group, these figures highlight the importance of the institutional environment [18]. Yorke and Longden also revealed multiple causes of dropout and concluded that due to this there is no singular solution on the problem of dropout neither for institutional nor for public policymakers [31].

### 2.2. Research into Background Factors of Dropout

Micro-level factors of dropout include students' socioeconomic backgrounds, gender, ethnic minority status, deficiencies rooted in their primary and secondary education, and academic results [10,11,24]. Research has shown that while women used to be less persistent, in recent decades, they have gained an advantage over men [32,33]. Since the expansion, there have been twice as many students enrolled in higher education from families of high socioeconomic status compared to those from low-status families; however, ten times as many from the former group have made it to graduation [12,34–36]. Other major causes are difficulties arising from the costs of studying and low economic status, which become risk factors due to the negative effects of doing paid work during studies and acquiring debt [28,37–39].

Searching for the institutional causes of attrition (also referred to as the campus effect) is a major research trend. The factors that contribute to large-scale attrition include institutional environment, relationship networks, and services provided by the institution, as well as institutional behaviour that is indifferent to or hinders student advancement for financial reasons. Furthermore, in some cases, the institutional culture and faculty attitude do not promote student integration and commitment [8,40–42]. When studying institutional environment, one must bear in mind that the character of an institution is

not equal to individual students' perceptions of it [40,43]. Another stumbling block is that the qualities of the environment are frequently confused with the intensity of student participation [7] or, in other words, integration [8]. In reality, the former refers to the context and the latter to the individual, so the two are to be separately analysed. Based on the available data, we intended to gain a better understanding of these phenomena by drawing a clear distinction between students' entry characteristics and institutional effects related to student integration and involvement.

### 2.3. The Effects of Institutional Integration

Early research into dropout turned its attention to the processes of student socialization, i.e., entering the institutions' relationship networks and moving away from external ones. Tinto [8,9] formulated his theory of student integration based on empirical evidence on student dropout. He distinguished three stages of student socialisation: declining and changing interactions with the student's previous environment, the transition of the prevailing normative system, and the acquisition of new patterns of interaction. All of these are essential for becoming a competent student. Students who leave their former community of family, friends, and school necessarily end up in some form of a "no man's land", i.e., in an anomic state [8]. During these three stages (separation, transition, and incorporation), students must first break away from the values and normative system of the former community, which does not yet entail the acquisition of a culture appropriate to the higher education environment and is inevitably accompanied by uncertainty. After successfully coping with the transition period, students can become integrated into higher education. However, if the transition is not smooth, they may not truly become higher education students, leading to dropout sooner or later [8].

While Astin's [44] model of student involvement argued that student persistence is best supported by students' active and time-intensive participation in the higher education institution, both in terms of extracurricular activities and paid or voluntary work on campus, Tinto [8] theorised that on-campus relationships play a crucial preventive role and that it is students' perception of their situation, influenced by academic contacts, that determines whether they become committed to completing their studies. Berger and Milem combined the concepts developed by Astin and Tinto into their own model of persistence and considered students' time investment and effort as a condition for persistence, as these lead to better student embeddedness and thus support student persistence [45,46]. Both Astin's [7] theory of involvement and Tinto's [8] theory of student integration hold that the more integrated students are into university life (by being members of on-campus groups and relationship networks and interacting with fellow students and lecturers), the more resistant to dropout they will become, because they acquire the norms and procedures that strengthen their attachment to the institution. The knowledge about the linkage between student integration into the institution's relationship network and dropout was further refined by research on non-traditional students, which detected the effect of perceptions of alienation and non-integration on the risk of dropout [34,47]. This line of research offered a possible explanation to successful integration, namely that children of highly educated parents can communicate better with academic staff and receive more recognition from their fellow students. Besides, students of higher financial status can spend more time at university as they are not strained by paid work.

Bean and Eaton [48] elaborated a more detailed model of the institutional environment, emphasizing the psychological aspects of the integration process. Kahu's [49] model embraced the structural and psychosocial factors measured on an institutional and individual level, which influences engagement and has an effect on the proximal and distal consequences, among them, the students' dropout. Kerby [50] extended Tinto's model and differentiated between the effects of external factors (such as national and educational climate), internal factors (such as institutional culture and climate), and adaptive factors (sense of place) on voluntary dropout.

In our research, we examined the impact of higher education communities on student perceptions [40]. Institutional environment can be regarded as a strong relationship network. We adopted the term embeddedness to describe students' attachment to this network [40]. Students' institutional intergenerational (e.g., with lecturers) and intragenerational (e.g., with fellow students) embeddedness can be measured by several dimensions, such as the orientation of relationships, the strength and multiplexity of embeddedness, the range of the individual's social activities inside and outside the institution and the relation between them. We found that, in contrast to early American research results [8], strong intragenerational embeddedness did not promote academic success but increased the likelihood of failure. We attributed this to institutional behaviour patterns and attitudes that did not support academic advancement [40]. On the other hand, we found that contact with academic staff had a significant positive effect on academic success and all of our studies confirmed that intergenerational institutional relationships remarkably supported student advancement [40].

In this study, we define intra-institutional relationships with lecturers as intergenerational and relationships with fellow students as intragenerational. The present study does not examine the effects of contact with parents and external friends, which has been explored in previous studies [51,52].

## 3. Research Questions and Hypothesis

The aim of this research was to explore the differences between the student groups on the persistence-dropout- scale in their sociodemographic composition and institutional integration. We compared students who dropped out (dropout students), students who are pursuing their studies but are uncertain about obtaining a degree (students at risk of dropout), and students committed to completing their studies (persistent students). Using bivariate methods, we compared the three student groups. We also employed multinomial logistic regression to examine the chances of dropping out, being at risk of dropout, and being persistent.

**Hypothesis 1 (H1).** *Based on the findings on the male disadvantage hypothesis (see [33]), we presume that the chance of dropout and the risk of dropout are larger among male students than female students.*

**Hypothesis 2 (H2).** *Based on our previous findings on the differences in social status with respect to graduation rates [40], we presume that the chance of dropout and the risk of dropout are larger among students from disadvantaged socioeconomic backgrounds.*

**Hypothesis 3 (H3).** *Based on findings on how institutional integration affects academic success [7,8], we presume that extensive relationships with academic staff and fellow students decrease the risk of dropout and increase the chance of being a persistent student.*

**Hypothesis 4 (H4).** *Based on our previous findings on dropout risk [24], we presume that the chance of dropout is larger among those who had to pay the tuition fee at least for one semester during studies.*

**Hypothesis 5 (H5).** *Based on [24], we presume persistence is adversely affected by paid work during studies.*

## 4. Materials and Methods

### 4.1. Databases

We based our research on the DEPART 2018 (*n* = 587) and PERSIST 2019 (*n* = 1895) survey databases of the CHERD-H (Center for Higher Educational Research and Development). The primary research objective was to explore the background and determinants of persistence and dropout. Both surveys were independent projects of the research centre, using a 20+ page questionnaire to gain information about present and former students' academic achievement, student careers, social background, multidimensional performance indicators (extra academic work, self-assessment, studies abroad, persistence, extracurricular activi-

ties, etc.), lifestyles, leisure activities, value preferences, forms of employment, motivations for study, and paid work, as well as dropout and institutional factors. The question blocks of the questionnaire had been regularly used and tested in our preceding research since the 2000s [40]. The questionnaire complied with the guidelines of data protection and voluntariness as laid down in the Code of Ethics of the University of Debrecen: respondents were informed both in writing and verbally that the completion of the questionnaire was voluntary and the results would be used anonymously for statistical analysis.

The DEPART 2018 survey targeted former students who had abandoned their studies without obtaining a degree in a Hungarian (mostly in the north-eastern region) institution of higher education in the previous ten years. Snowball sampling was used to recruit the sample group. As a result, the sample is not representative; yet, considering that this is a unique and hidden group, which is difficult to reach from higher education networks, such a (relatively) high sample size is a rarity.

The PERSIST 2019 survey was conducted in 2018–2019 among higher education students in the same north-eastern region of Hungary and in mainly Hungarian minority institutions of four regions in the neighbouring countries. For our present study, we excluded the subsample taken abroad and examined students only from the higher education institutions of Eastern Hungary ($n = 857$), as this is the region where the proportion of disadvantaged and non-traditional students who are at risk of dropout is the highest in the country [53]. We used the PERSIST database as a reference subsample for which to compare dropout students. The sample taken in Hungary is a quota sample representative for university faculties, areas of study, and financing.

### 4.2. Examined Variables and Instruments

To measure persistence, we used the statements about commitment to graduation from the Persistence/Voluntary Dropout Decision Scale [54], Cronbach $\alpha = 0.818$. The nine-item list of questions explores students' commitment to graduation, how useful they find their studies, and what efforts they make to meet class and exam requirements [55,56]. In the PERSIST 2019 database, we created a principal component from the scale (KMO index = 0.846, explained variance: 59.421 percent) and calculated its average. Students for whom the indicator was below the average were termed students at risk of dropout and those with an above-average value were characterized as persistent students. For this analysis, the DEPART 2018 database and the Hungarian subsample of the PERSIST 2019 database were merged ($n = 1441$) and the participants were divided into three groups: dropout students ($n = 584$), students at risk of dropout ($n = 453$), and persistent students ($n = 404$). As international studies mostly focus on the comparison of persistent students and non-persistent students or graduates and dropout students [57], our unique approach of comparing three groups on the persistence-dropout scale is a methodological novelty in European research.

Our first explanatory variable was the sex of the respondent (1—male, 0—female). The questionnaire contained several questions about respondents' social background. From the variables, the following ones were examined in multiple ways: parents' educational attainment (1—tertiary, 0—secondary or primary), place of residence at 14 years of age (1—big city, 0—village or small town), respondents' financial status (a composite index indicating the possession of durable goods (1–6)). Components of the index: Does the respondent possess an apartment or house, a car, an above-average smartphone (e.g., iPhone), an above-average computer or laptop, a tablet or e-book reader, and savings for house purchase? A subjective indicator of individual financial situation explores whether the respondent can afford a significant purchase or is unable to cover even the basic expenses (1–4): (1) Often I do not have enough money for basic everyday necessities. (2) Sometimes I do not have enough money for everyday expenditures. (3) I have everything I need but cannot afford larger expenditures. (4) I have everything I need and can also afford larger expenditures. Additionally, the relative financial status was measured on a 1–5 scale, where 3 is the average situation.

Based on the literature, we also examined the effects of some risk factors in higher education training and experience. We asked students about the number of semesters they had spent on the tuition-paying program (coded as 0—none, 1—at least one semester) and whether they had done any paid work and how frequently (0—never, 1—at least once a year). We measured intergenerational institutional embeddedness by asking whether there were any academic staff with which they could discuss private, academic, or political topics, turn to for advice, or who paid special attention to their careers (answer options: one, more than one, none; composite index coded 0–16). The same questions were asked with respect to fellow students to measure intragenerational embeddedness (answer options: yes or no; composite index coded 0–10).

*4.3. Methods*

We carried out the analysis of the data with SPSS 24. First, we conducted bivariate analysis to compare the three student groups (persistent, at risk of dropout, and dropped out). We applied contingency tables and a chi-squared test for categorical (dummy) explanatory variables, namely students' gender, mothers' and fathers' level of education, place of residence at the age of 14, whether student has spent at least one semester in a self-funded programme, and whether student has done paid work during their studies. We conducted analysis of variance for continuous explanatory variables (objective, subjective, and relative financial status and relationships with academic staff and fellow students). At the second stage, we carried out multinomial logistic regression analysis to explore the effects of explanatory variables on the chances of being a persistent student, being at risk of dropout, and dropping out.

## 5. Results

*5.1. Bivariate Analysis*

The main question of this study is what differences can be detected between persistent students, students at risk of dropout, and dropout students in their sociodemographic backgrounds and the characteristics of their higher education studies. We found significant differences across the three groups for nearly all of the examined variables (see Tables 1 and 2). Only the place of residence at the age of 14 did not differ across the three groups (in the total sample, 65.7% of students lived in small towns or villages and 34.3% in larger towns or cities).

Tables 1 and 2 show that males are underrepresented among persistent students and overrepresented among both students at risk of dropout and those who dropped out. The share of mothers with a tertiary degree is about two-thirds among persistent students, and is slightly smaller among students at risk of dropout but only about one-third among dropout students. The pattern is similar for graduate fathers, albeit with lower proportions (45.2%, 40.6%, and 22.8%, respectively). This is likely due to the fact that, in Hungary, men were traditionally steered towards vocational education to become employed and start earning sooner, while women were often encouraged to attend general secondary education after which they were more likely to enrol in higher education. In line with previous results, objective, subjective, and relative financial situation is the best among persistent students, somewhat worse among students at risk of dropout, and the worst among those who dropped out. Concerning institutional integration, the relationship with fellow students is the most extensive in the persistent group and the poorest among dropout students but only the persistent group has a relatively high average in terms of the relationship with academic staff, as the dropout group and the students at risk of dropout do not differ in this area. A further result is that about one-half of dropout students had to pay tuition for at least one semester whereas, in the other two groups, this proportion is only about one-third. Finally, it is an interesting finding that the share of those who did paid work during their studies is the largest among those at risk of dropout and the smallest among dropout students.

**Table 1.** Results of the relationship between categorical explanatory variables and the three groups by dropout status (contingency tables, chi-squared test; for underlined values the adjusted residual is larger than two).

| | Persistent | At Risk of Dropout | Dropped Out | Total | chi-Squared Sign | *n* |
|---|---|---|---|---|---|---|
| **Male** | 32.8% | 47.1% | 46.2% | 42.6% | 0.000 | 1373 |
| **Tertiary level education (mothers)** | 64.1% | 59.6% | 31.5% | 49.5% | 0.000 | 1441 |
| **Tertiary level education (fathers)** | 45.5% | 40.6% | 22.8% | 34.8% | 0.000 | 1441 |
| **Ever participated in the tuition-paying program** | 31.4% | 36.0% | 50.3% | 40.0% | 0.000 | 1374 |
| **Did paid work** | 57.8% | 64.1% | 52.1% | 57.6% | 0.001 | 1387 |

Reference values: female; completed primary or secondary education; only studied tuition-free; did no paid work during studies.

**Table 2.** Results of the relationship between continuous explanatory variables and the three groups by dropout status (comparison of means, standard deviation in brackets, significance of ANOVA).

| | Persistent | At risk of Dropout | Dropped Out | Total | ANOVA Sign | *n* |
|---|---|---|---|---|---|---|
| **Objective financial situation index (1–6)** | 1.76 (1.43) | 1.63 (1.4) | 1.05 (1.24) | 1.43 (1.38) | 0.000 | 1441 |
| **Subjective financial situation (1–4)** | 3.3 (0.56) | 3.17 (0.59) | 2.9 (0.75) | 3.1 (0.67) | 0.000 | 1389 |
| **Relative financial situation (1–5)** | 3.3 (0.69) | 3.22 (0.72) | 2.93 (0.85) | 3.13 (0.78) | 0.000 | 1389 |
| **Relationships with academic staff, index (0–16)** | 3.6 (3.5) | 2.58 (3.17) | 2.52 (3.17) | 2.84 (3.3) | 0.000 | 1441 |
| **Relationships with fellow students, index (0–10)** | 8.14 (2.18) | 7.58 (2.54) | 6.43 (3.31) | 7.27 (2.89) | 0.000 | 1441 |

*5.2. Multinomial Logistic Regression Results*

In this section, we use multinomial logistic regression to examine the effect of sociodemographic and institutional factors on the chances of dropping out, facing the risk of dropout, and being persistent. Based on Table 3, the chance of dropping out compared to being persistent is larger if the student is male or lived in large town or city at the age of 14. Although urban students from unfavourable social backgrounds are more likely to enrol in higher education, they may not be as valued as rural students. Rural students may be more committed, have a different value system, a different work ethic, an aspirational nature, a desire for social mobility, and a sense of duty, which may ultimately lead to greater persistence. In addition, the type of settlement of origin may be related to the chosen field of education and thus also affect student persistence. Moreover, the chance of dropping out compared to being persistent is larger if the student has a disadvantaged social background (i.e., the mother does not have tertiary education, and the objective, subjective, and relative financial situation is below average). Furthermore, the chance of dropping out relative to persistence is also higher among those who have poor relationships with academic staff and fellow students, those who paid tuition for at least one semester, and those who did not do paid work during their studies. It was only the fathers' educational attainment that did not exert an effect.

When examining the chance of being at risk of dropout compared to being persistent, much fewer effects can be detected (Table 4). This chance is also larger for male students but social background has a relatively small effect: an unfavourable subjective financial situation elevates the risk of dropout, while other social background indicators have no impact. However, institutional integration also matters in this case, with those who have poor relationships with academic staff and fellow students being more at risk of dropout.

**Table 3.** Multinomial logistic regression results on the factors affecting the chance of dropping out compared to being persistent (Nagelkerke $R^2$ = 0.243).

| | B | Std. Error | Wald | df | Sig. | Exp(B) | 95% Confidence Interval for Exp(B) | |
|---|---|---|---|---|---|---|---|---|
| | | | | | | | Lower Bound | Upper Bound |
| Constant | 4.528 | 0.547 | 68.606 | 1 | 0.000 | | | |
| Gender (1:male) | 0.713 | 0.162 | 19.309 | 1 | 0.000 | 2.039 | 1.484 | 2.802 |
| Mother's educational attainment (1: tertiary) | −0.990 | 0.175 | 32.071 | 1 | 0.000 | 0.372 | 0.264 | 0.523 |
| Father's educational attainment (1: tertiary) | −0.275 | 0.188 | 2.141 | 1 | 0.143 | 0.760 | 0.526 | 1.098 |
| Objective financial status index | −0.214 | 0.064 | 11.238 | 1 | 0.001 | 0.807 | 0.713 | 0.915 |
| Subjective financial status | −0.635 | 0.140 | 20.436 | 1 | 0.000 | 0.530 | 0.402 | 0.698 |
| Relative financial status | −0.258 | 0.118 | 4.746 | 1 | 0.029 | 0.773 | 0.613 | 0.974 |
| Place of residence at the age of 14 (1: big city) | 0.367 | 0.169 | 4.740 | 1 | 0.029 | 1.444 | 1.037 | 2.009 |
| Relationships with academic staff | −0.060 | 0.024 | 6.433 | 1 | 0.011 | 0.942 | 0.899 | 0.986 |
| Relationships with fellow students | −0.136 | 0.032 | 17.909 | 1 | 0.000 | 0.872 | 0.819 | 0.929 |
| Participation in the tuition-paying program | 0.865 | 0.160 | 29.083 | 1 | 0.000 | 2.374 | 1.734 | 3.251 |
| Doing paid work | −0.368 | 0.159 | 5.371 | 1 | 0.020 | 0.692 | 0.507 | 0.945 |

**Table 4.** Multinomial logistic regression results on the factors affecting the chance of being at risk of dropout compared being persistent (Nagelkerke $R^2$ = 0.243).

| | B | Std. Error | Wald | df | Sig. | Exp(B) | 95% Confidence Interval for Exp(B) | |
|---|---|---|---|---|---|---|---|---|
| | | | | | | | Lower Bound | Upper Bound |
| Constant | 1.675 | 0.537 | 9.727 | 1 | 0.002 | | | |
| Gender (1: male) | 0.687 | 0.151 | 20.614 | 1 | 0.000 | 1.988 | 1.478 | 2.675 |
| Mother's educational attainment (1: tertiary) | −0.134 | 0.167 | 0.648 | 1 | 0.421 | 0.874 | 0.630 | 1.213 |
| Father's educational attainment (1: tertiary) | −0.118 | 0.171 | 0.474 | 1 | 0.491 | 0.889 | 0.636 | 1.243 |
| Objective financial status index | −0.011 | 0.057 | 0.039 | 1 | 0.844 | 0.989 | 0.885 | 1.105 |
| Subjective financial status | −0.317 | 0.136 | 5.389 | 1 | 0.020 | 0.728 | 0.558 | 00.952 |
| Relative financial status | −0.035 | 0.112 | 0.100 | 1 | 0.752 | 0.965 | 0.775 | 1.202 |
| Place of residence at the age of 14 (1: big city) | 0.187 | 0.157 | 1.420 | 1 | 0.233 | 1.206 | 0.886 | 1.641 |
| Relationships with academic staff | −0.092 | 0.023 | 16.667 | 1 | 0.000 | 0.912 | 0.872 | 0.953 |
| Relationships with fellow students | −0.069 | 0.032 | 4.768 | 1 | 0.029 | 0.933 | 0.878 | 0.993 |
| Participation in the tuition-paying program | 0.180 | 0.154 | 1.358 | 1 | 0.244 | 1.197 | 0.885 | 1.618 |
| Doing paid work | 0.291 | 0.150 | 3.766 | 1 | 0.052 | 1.338 | 0.997 | 1.796 |

As Table 5 shows, the chance of dropping out compared to being at risk of dropout is larger if students have a disadvantaged social background (i.e., the mother does not have tertiary education and the objective, subjective, and relative financial situation is below average) and is also higher among those who paid tuition for at least one semester and did not do paid work during their studies. Students' gender and former place of residence are not important in this case. As for institutional integration, a poor relationship with fellow students has an impact but relationships with academic staff do not have an effect.

In summary, our findings show that both the chance of dropout and the risk of dropout are larger among male students, those with an unfavourable subjective financial situation, and those who have poor relationships with academic staff and fellow students. Furthermore, the chance of dropout is increased among students who had to pay tuition for at least one semester and did no paid work during their studies. We have also found that the effect of social background is larger on the chance of dropping out compared to being persistent than on the chance of being at risk of dropout compared to being persistent but institutional integration indicators have a sizeable impact on both.

**Table 5.** Multinomial logistic regression results on the factors affecting the chance of dropping out compared to being at risk of dropout (Nagelkerke $R^2$ = 0.243).

| | B | Std. Error | Wald | df | Sig. | Exp(B) | 95% Confidence Interval for Exp(B) | |
|---|---|---|---|---|---|---|---|---|
| | | | | | | | Lower Bound | Upper Bound |
| Constant | 2.853 | 0.466 | 37.517 | 1 | 0.000 | | | |
| Gender (1: male) | 0.025 | 0.147 | 0.030 | 1 | 0.863 | 1.026 | 0.769 | 1.367 |
| Mother's educational attainment (1: tertiary) | −0.855 | 0.161 | 28.193 | 1 | 0.000 | 0.425 | 0.310 | 0.583 |
| Father's educational attainment (1: tertiary) | −0.157 | 0.177 | 0.787 | 1 | 0.375 | 0.855 | 0.604 | 1.209 |
| Objective financial status index | −0.203 | 0.060 | 11.302 | 1 | 0.001 | 0.816 | 0.725 | 0.919 |
| Subjective financial status | −0.318 | 0.123 | 6.659 | 1 | 0.010 | 0.727 | 0.571 | 0.926 |
| Relative financial status | −0.222 | 0.107 | 4.279 | 1 | 0.039 | 0.801 | 0.649 | 0.988 |
| Place of residence at the age of 14 (1: big city) | 0.180 | 0.155 | 1.341 | 1 | 0.247 | 1.197 | 0.883 | 1.622 |
| Relationships with academic staff | 0.032 | 0.024 | 1.808 | 1 | 0.179 | 1.033 | 0.985 | 1.082 |
| Relationships with fellow students | −0.068 | 0.027 | 6.090 | 1 | 0.014 | 0.935 | 0.886 | 0.986 |
| Participation in the tuition-paying program | 0.685 | 0.147 | 21.757 | 1 | 0.000 | 1.984 | 1.488 | 2.646 |
| Doing paid work | −0.659 | 0.150 | 19.316 | 1 | 0.000 | 0.517 | 0.386 | 0.694 |

## 6. Discussion

Our study explored sociodemographic and institutional factors affecting dropout and persistence. Recent trends in higher education (a massive expansion and its deceleration and the introduction and challenges of the multi-cycle system) have increased dropout both in Hungary and internationally in the last decades.

Based on our results, the chance of dropout and the risk of dropout relative to being a persistent student are about doubled among male students compared to female students, which is in accordance with our first hypothesis. Our results support the findings of earlier studies that had registered better outcomes and greater persistence for women [10,33,58].

In accordance with our previous findings and our second hypothesis, the chance of dropout is larger among students with a disadvantaged socioeconomic background. Our results highlight the disadvantage of first-generation higher education entrants and of students of low financial status. The lack of parents' higher education experience and of cultural capital in the family (refined language use, reading, and abstract thinking) are enough to put these students at a severe disadvantage in managing their studies and exams. Additionally, the unfavourable or below-average financial situation that low parental education is usually accompanied by can hinder the payment of the tuition fee and is more likely to drive students to terminate their studies. Social inequalities also manifest themselves in higher education dropout: low cultural and economic capital dramatically increase the chance of dropout, whereas a mother with a tertiary degree enhances the chance of persistence. Consequently, our findings support the view that higher education perpetuates, rather than diminishes, social inequality, and therefore students of low social status are much more likely to terminate their studies without graduation than their higher-status peers [35,36].

According to our findings, nearly all social background indicators affect the chance of dropout compared to being a persistent student, with the exception of the fathers' educational attainment. However, the chance of being at risk of dropout compared to being persistent is increased by only one indicator, namely the unfavourable subjective financial status. In other words, the negative effect of a disadvantaged social background can be primarily detected on actual dropout and not on the risk of dropout. This is one of the novel findings of our study.

However, institutional integration has an effect in both cases (the chance of being at risk of dropout and the chance of dropout). In accordance with our third hypothesis, extensive relationships with academic staff and fellow students decrease the risk of dropout and increase the chance of being a persistent student. As formulated by the theory of student integration, if a student has a peer to share their academic workload with, to ask for help with their studies, and to rely on in other areas of life, that is, if a student is

well-integrated, it serves as a significant protective factor. This confirms the validity of Tinto's [8] original model of integration in today's higher education, as students' intragenerational relationships with fellow students both safeguard against dropout and strengthen persistence. In line with Pascarella and Terenzini's [54] concept, which may further refine Tinto's model, our data proved that contact with academic staff also increased persistence. Support and personal attention from academic staff can be of great help to overcome academic difficulties and, ultimately, they pave the way for successful graduation. In light of our findings, the sociocultural and institutional characteristics of dropout students and persistent students seem to reflect the Matthew effect: the disadvantaged social background of dropout students is coupled with a limited institutional relationship network, so many of them are left alone with their academic problems, whereas persistent students do not only come from favourable backgrounds but their institutional embeddedness also contributes to their successful graduation.

In accordance with our fourth hypothesis, the chance of dropout compared to being persistent or being at risk of dropout is nearly two times larger among those who had to pay tuition even for one semester during studies. Our findings clearly disprove educational policymakers' conviction that paying tuition for one's studies has an educative function as it can result in a more responsible academic attitude in students. Our survey showed that participation in the tuition-paying training program for at least one semester was a primary risk factor of dropout. Controlled by all other variables, it also reduced persistence as tuition-paying students were less certain about successful graduation. Having looked at tuition-paying students' fields of study and having conducted interviews with them, we found that persistent students in this group were doing courses mostly in the fields of economics or social sciences, where tuition fees were relatively low. They had a stable financial background and opted for the tuition-paying training program of their own accord. Tuition-paying students who had dropped out or who were at risk of dropout began their studies in the state-funded program and had to start paying tuition later because of their academic results. Originally, they did not undertake the risk of paying. We also found that the chance of being at risk of dropout compared to being persistent was not increased by this factor, so participating in a tuition-paying programme is another factor, which increases dropout but does not affect the risk of dropout.

Finally, contrary to our earlier findings [28] and our fifth hypothesis, this study identified a positive effect of doing paid work on persistence. Those who did paid work during studies have a smaller chance of dropping out compared to being at risk of dropout or being persistent. This may be due to the fact that paid employment has recently become more common among students in CEE countries. The focus of higher education has shifted from intellectual to professional training, students have become more utilitarian, and lecturers have grown more tolerant of absences due to employment. Furthermore, Hungarian higher education has seen an increase in the number of internships in parallel with the introduction of dual training, whereby students spend some time working as part of their curriculum. All this may have led students to do paid work to become more enthusiastic about their studies because they experience the job prospects first hand, which makes them more likely to be persistent. The background of this phenomenon could also be Astin's [44] argument that on-campus work clearly increases persistence compared to off-campus work and another finding [59] that persistence is strongly dependent on the intentionality and regularity of work, which we did not investigate in this study.

One of the limitations of this study is that the sample and the region included in the survey do not cover the entire country. Thus, our results are not applicable to Hungary in general but only to a specific region of Eastern Hungary, which can be regarded as a disadvantaged region by the EU standards of GDP per capita and population size [60]. Due to the COVID-19 pandemic, there has been no large-scale data collection in this region since 2019, so we cannot yet observe the impact of the pandemic, nor that of the war in Ukraine. A further limitation is the manner of sampling, especially the snowball method, which we used among dropout students because of their limited availability. What is

missing from this paper is the comparison of results by fields of study and the discussion of the relationship between various fields of study and dropout, persistence, and their determinants. Future research should explore these questions in more depth.

## 7. Conclusions

One of our research questions concerns whether academic and social integration contribute to students' commitment to completing their studies more than the social status of students' families does. Our conclusion is that it is poor institutional integration, which primarily reduces the persistence of students, as it was pointed out in former research (see [8,9,44,46,50]). However, our findings reveal that in addition to institutional embeddedness, actual dropout is also greatly determined by social background, which is a novel result of our research. This could be due to our methodological approach, simultaneously examining the three groups (persistent students, students at risk of dropout, and dropout students) and controlling for the effects of integration variables with social background, which cannot be found in the literature.

Our results are also novel from an education policy perspective, as most higher education institutions can, at best, only identify students at risk of dropping out and have little information on students who already dropped out. Our results suggest that, in order to reduce dropout, higher education policy needs to simultaneously reduce the negative effects of unfavourable social and financial backgrounds and increase students' institutional embeddedness. There is a need for multiple policies to reduce dropout, as Yorke and Longden [31] suggested.

Higher education funding in the region should support students' education and living expenses while also enhancing their professional socialization and embeddedness at university through academic or practical efforts.

**Author Contributions:** Conceptualization, G.P.; formal analysis, H.F. and K.K.; methodology, H.F. and K.K.; software, H.F. and K.K.; validation, G.P., H.F. and K.K.; writing—original draft, G.P., H.F. and K.K.; writing—review and editing, G.P., H.F. and K.K. All authors have read and agreed to the published version of the manuscript.

**Funding:** The research on which this paper is based has been implemented by the MTA-KFK 2021-2025/MTA-DE-Parent-Teacher Cooperation Research Group and with the support provided by the Research Programme for Public Education Development of the Hungarian Academy of Sciences.

**Institutional Review Board Statement:** This research was conducted in accordance with the Declaration of Helsinki. The ethical committee of the University of Debrecen approved this study. The research was conducted ethically, the results are reported honestly, the submitted work is original and not (self-)plagiarized, and authorship reflects the individuals' contributions.

**Informed Consent Statement:** Informed consent was not required for the study.

**Data Availability Statement:** Data are available only on request due to ethical restrictions.

**Conflicts of Interest:** The authors declare no conflict of interest.

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
