# Peer review of "Factors Influencing the Chance of Dropout or Being at Risk of Dropout in Higher Education"

_education, doi:10.3390/educsci12110804_

Round 1
Reviewer 1 Report
Dear Authors,
Please see my comments attached.
Kind regards,
The reviewer

Author Response
We made an English language proofreading of our paper by a native speaker; we hope that there are no mistakes any more. In the introduction, we put more references and deleted one sentence. In the literature review, we put the suggested five references about our topic. Thanks for the suggestions.
Reviewer 2 Report
The article addresses a very interesting topic and although there are previous studies that talk about it, the authors approach the problem from a novel perspective by analyzing not only students at risk of dropping out but also those who have already done so. The aim of this paper is to establish what sociodemographic and institutional factors cause students to drop out, doubt their intentions of obtaining a degree, or move confidently toward fulfilling their ambitions. In addition to a number of hypotheses that are contrasted throughout the paper, such as male disadvantage in relation to dropout, higher dropout probabilities among students from disadvantaged socioeconomic backgrounds, institutional integration as a determinant of academic success, or paid work as another possible cause of this. They use statistical methods such as bivariate analysis and regression to examine dropout probabilities, dropout risk and persistence for three groups of students.
Both the theoretical and empirical analyses meet the requirements of a scientific paper. The methodology is correct and the state of the question is well argued and substantiated. Despite some limitations of the study, such as the sample size or the fact that the region included in the survey does not cover the whole country, as they themselves comment in the results section (line 467), I consider that the work meets the requirements to be published in the journal.
Author Response
He/she accepted our paper without changes. Thank you for your support!
Reviewer 3 Report
Dear Authors,
First of all, I congratulate you for working on a topic of such relevance at the European level, and which is of vital importance from the point of view of social justice. After an in-depth reading of your manuscript, it has been decided to accept the manuscript, subject to a series of modifications that I am including below:
- Firstly, they use the term school dropout , however, the literature has criticized this term for implying a simplistic conception of the phenomenon, replacing it with early school leaving, or early dropout from education and training. I encourage you to review this aspect.
- An important part of the hypotheses presented have already been extensively tested in the literature, so it would be important for you to reaffirm the extent to which your study is innovative. This should also be clear in the discussion.
Best wishes,
Reviewer
Author Response
In the conceptualization part, we included the suggested term: early school leaving, which is used in primary and secondary education instead of dropout. Based on the suggestions of the opponent, in the conclusion, and in the abstract we emphasized the novelty of our findings more. Thank you for your work and suggestions!
Round 2
Reviewer 1 Report
Dear Authors,
The manuscript has been substantially improved and has now the quality to be published in Education Sciences.
Regards,
The Reviewer
Reviewer 3 Report
All my concerns have been taken into account